# Why do children attend school, engage in other activities or socialise when they have symptoms of an infectious illness? A cross-sectional survey

Lisa Woodland [ORCID],[1] Louise E Smith [ORCID],[1] Rebecca K Webster [ORCID],[2] Richard Amlôt,[3] James G Rubin[1]

¹Psychological Medicine, King's College London, London, UK
²Psychology, The University of Sheffield, Sheffield, UK
³Behavioural Science and Insights Unit (BSIU), UK Health Security Agency, London, UK

**Correspondence to**
Lisa Woodland;
lisa.woodland@kcl.ac.uk

## ABSTRACT

**Objectives** To prevent the spread of infectious disease, children are typically asked not to attend school, clubs or other activities, or socialise with others while they have specific symptoms. Despite this, many children continue to participate in these activities while symptomatic.

**Design and setting** We commissioned a national cross-sectional survey with data collected between 19 November and 18 December 2021.

**Participants** Eligible parents (n=941) were between 18 and 75 years of age, lived in the UK and had at least one child aged between 4 and 17 years. Parents were recruited from a pre-existing pool of potential respondents who had already expressed an interest in receiving market research surveys.

**Outcome measures** Parents were asked whether their children had exhibited either recent vomiting, diarrhoea, high temperature/fever, a new continuous cough, a loss or change to their sense of taste or smell in the absence of a negative (PCR) COVID-19 test ('stay-at-home symptoms') since September 2021 and whether they attended school, engaged in other activities outside the home or socialised with members of another household while symptomatic ('non-adherent'). We also measured parent's demographics and attitudes about illness.

**Results** One-third (33%, n=84/251, 95% CI: 28% to 39%) of children were 'non-adherent' in that they had attended activities outside the home or socialised when they had stay-at-home symptoms. Children were significantly more likely to be non-adherent when parents were aged 45 and younger; they allowed their children to make their own decisions about school attendance; they agreed that their child should go to school if they took over-the-counter medication; or they believed that children should go to school if they have mild symptoms of illness.

**Conclusion** To reduce the risk of spreading disease, parents and teenagers need guidance to help them make informed decisions about engaging in activities and socialising with others while unwell.

## INTRODUCTION

In order to reduce the spread of infectious illness within schools, children who have specific symptoms, including fever, diarrhoea and vomiting, are commonly advised to remain at home.[1 2] While this has been the case for many years, this message was made more urgent by the COVID-19 pandemic. In the UK, throughout most of 2020 and 2021, anyone with a new continuous cough, a high temperature or a loss or change to their sense of taste or smell were asked not to attend school or work nor to interact with people outside their household unless they had a negative COVID-19 PCR test result.[3 4] This message is also important for other symptoms of infectious illnesses. For instance, in February 2022, cases of norovirus were 48% higher than expected in educational settings,[5] resulting in a warning to the public about the health threat.[5] Children are particularly susceptible to infections because their immune systems are developing,[6] they are often in close contact with other children, and although they can adhere to hygiene practices,[7] they are at risk of infections.[2 8 9] Preventing ill children from mixing with others is essential in mitigating the spread of infection.[10 11] The UK Government has issued guidance on how long a child should not attend school for when they show symptoms of an infectious illness.[2]

A systematic review indicates that a large proportion of symptomatic children may be attending school (known as 'school-based presenteeism').[12] For example, one survey of 3040 secondary school pupils in Norway reported that 58% had attended school in

the past year despite feeling so ill that they should have stayed at home.[13] One in six British parents reported that they would send their child to school even if they were currently experiencing diarrhoea or vomiting.[14] The systematic review identified five themes that were shown to impact school-based presenteeism: children's characteristics, children's and parents' motivations and attitudes towards school; organisational factors and school sickness policy.[12] Another systematic review found similar findings and suggested that people are more likely to attend school or work with symptoms of an infectious illness if they: are unsure of the sickness guidelines; are worried about disciplinary action; are unable to find alternative child care; are concerned about their workload; feel they have missed too much work; perceive a culture of presenteeism in the organisation; and perceive their illness to be mild or non-infectious.[15] The studies included in the systematic review focus only on work-based or school-based presenteeism and the findings in relation to children in the UK are limited. However, hypotheses can be drawn; and children who have positive motivations towards school, worry about school attendance and when the symptoms of illness are mild, will be more at risk of non-adherence.

## STRENGTHS AND LIMITATIONS OF THIS STUDY

⇒ A strength of the study is that we investigated reported behaviour, rather than intentions.
⇒ The study used a non-probability sample, thus there is a risk of some sampling bias.
⇒ Data about children's demographics were not collected, which limits our ability to interpret the data.

In this study, we aimed to identify the proportion of children who were non-adherent (eg, attended school, clubs or other activities or who socialised with people outside their household) while they had symptoms that should require them to remain at home. We also aimed to identify the possible parent-level and child-level risk factors for engaging in these activities. The results of which can then be used to inform interventions that target these risk factors as a means to prevent the spread of infectious diseases.

## METHODS
### Design
We used data from a longitudinal study about perceptions of COVID-19, trust in public officials and other health attitudes and behaviours. We commissioned Ipsos MORI[16] to conduct this national cross-sectional online survey of 5000 participants at each wave. This paper is based on data from wave 5, which collected data between 19 November and 18 December 2021.

### Participants
Non-probability sampling was used for ease of participant recruitment and to limit study costs. Participants were recruited from a pre-existing pool of potential respondents who had already expressed an interest in receiving market research surveys. Participants were recruited by advertising via platforms such as social media, online gaming sites and news outlets. In Europe, 38% of the volume is recruited through social media, 26% through self-recruitment and referral and 36% through affiliate networks and media agencies. Further information about

the company's recruitment process can be found in their guidance.[17]

Participants aged between 18 and 75 years, who lived in the UK, were eligible for the survey. Quotas were set on the main sample: age interlocked by gender; government office region; working status; and social grade based on PAMCo V.4 (October 2018 to September 2019) data.[18] One per cent of participants were removed due to Ipsos MORI's quality control procedures.[17] The survey company performs automated quality checks, such as country geo-IP validation and removing duplicate email identifications. Further checks were made on the final dataset, and data that indicated inconsistencies, such as straight-lining, when the participant selects the same response throughout the survey, and incomplete surveys were removed.

Recruitment followed a complex pattern, in which responders from earlier waves were invited to participate and additional respondents from the market research panel were then invited to take part to take the place of non-responders. Response rates are not provided because they are not an accurate indicator of bias in quota samples—given that the underlying sample frame consists of people who have previously self-selected to receive invitations to take part in surveys, understanding whether the achieved sample (within each quota) is representative of the sample frame provides no information as to whether it is also representative of the wider general population.

Participants were paid between €1 and €1.50 for completing the survey.

The target population was parents of dependent children. As this was a subsample of a wider survey, the participants and sample size were based on the needs for the main study, which set quotas to obtain a sample broadly representative of the UK adult population. In our parent sample: 80% were employed, in the UK 69% of parents are employed in lone parent households and both parents are employed in 74% of couple households[19]; 54% earned £35 000 and over and the average disposable income for UK households with children was £35 049[20]; and 49% of parents were highly educated, which is higher than the 34% of parents and non-parents in England and Wales.[21]

### Study materials
The wording of all survey items is available in the online supplemental file 1.

### Participant demographics
We asked parents to report their gender, age, postcode (to derive region), household income, employment status, ethnicity and level of education. A non-response option was available for each of these demographic questions ('prefer not to say', 'prefer not to answer' and for gender only, parents could also respond 'in another way'). Due to space limitations in the survey, we did not collect child demographic data.

### Identification of children with symptoms of illness

Parents were asked to consider their four (or fewer) youngest children aged between 4 and 17 years and to report for each child any symptoms they had experienced since the start of the school year (September 2021). We restricted the number of children parents could report on to four to allow most parents to report the symptoms for all of their children, while limiting the length of the survey. In Scotland, the academic year started in August 2021, therefore, participants in Scotland were asked about their child's symptoms since '*about* the start of the school year (September 2021)'. If a child had experienced multiple bouts of illness during that period, we asked parents to report the one set of symptoms they perceived as most severe. Parents were asked to report all symptoms that applied to their child from a list of 14 symptoms of infectious illnesses and ailments common in children, listed by the UK Government.[2]

### Children's activities while symptomatic

Parents who had at least one symptomatic child were asked to consider the child who most recently exhibited symptoms and to report whether, when they had symptoms, they engaged in any of the following activities (excluding online): going to school; going to a club or lesson outside of school; visiting someone from another household; having someone from another household visit them; having someone from another household visit the household in general. Parents were asked to 'tick any (activity) that applied', or they could respond 'none of these' or 'prefer not to say'.

We also asked parents whether their child had taken a COVID-19 test (lateral flow test (LFT) test or a PCR test) while symptomatic, and if so, the result of the test.

### Parent attitudes about their child and perceptions about illness

We asked parents 13 statements about their attitudes concerning their child (eg, my child has missed too much school since September this year) and about common perceptions about illness (eg, other children with common illnesses (eg, a cold) go to school). We asked parents to respond to each statement on a five-point Likert scale from 'strongly agree' to 'strongly disagree'; they could also respond 'don't know' or 'prefer not to say'.

### Analysis

Analysis was conducted in SPSS, Version 27.[22] We created binary and multinominal variables for parent demographics (presented in table 1).

We recoded symptoms that necessitated the child to 'stay at home' according to Government guidance as being: a new, continuous cough or a loss or change to their sense of taste or smell in the absence of a negative PCR result; a high temperature; vomiting; and diarrhoea. We restricted our analyses to parents of a child who had exhibited one or more of these stay-at-home symptoms.

We created a single binary variable indicating whether the child had attended at least one activity outside the home (except to get a PCR or LFT COVID-19 test), which included any interaction (inside or outside the home) with someone from another household ('non-adherence'). Children who had stayed at home (except to get a PCR or LFT COVID-19 test) and had not interacted with someone from another household were categorised as adherent.

For all variables, we coded the responses 'in another way', 'prefer not to say/answer' and 'don't know' as missing data. Participants had to complete each question before moving onto the next; as such, there were no other missing data.

We ran separate binary logistic regressions to test univariable associations between children's adherence and: parents' demographic variables, attitudes about their child and perceptions about illness (as continuous variables). We ran a second set of binary logistic regressions adjusting for parent gender, age, region, income, employment status and education level. The potential cofounders were entered into the regression model at the same time for every test.

We also reran sensitivity analyses investigating associations with school attendance only (excluding all other out-of-home activities and social interactions). These analyses were to identify differences in associations between parents of children who were non-adherent in any behaviour compared with children who were non-adherent due to only attending school.

### Reporting

We used the Strengthening the Reporting of Observational Studies in Epidemiology cross-sectional checklist when writing this manuscript.[23]

### Patient and public involvement

None.

## RESULTS

Overall, 4962 participants completed the main survey, of whom 941 (19%) indicated that they had a child aged between 4 and 17 years. This proportion of respondents is as expected, as 22% of households in the UK have a dependent child.[24]

Of these 941 parents, 251 reported that their child had experienced symptom(s) that required them to stay at home.

Parents (n=941) reported on symptom(s) for 1533 children aged 4–17 years. Overall, 48% of parents (n=454, 95% CI: 45% to 51%) reported that at least one of their children had experienced at least one symptom and 27% (n=251, 95% CI: 24% to 30%) reported that at least one child had experienced at least one stay-at-home symptom. Of the 251 children who had at least one stay-at-home

**Table 1** Associations between parent demographics and attitudes, and whether their child (age 4–17 years, with at least one stay-at-home symptom) was non-adherent (n=251)

| Parent demographics and attitudes about sending children to school while symptomatic | Level | Children were adherent (%) | Children were non-adherent (%) | Odds ratio (95% CI) | P value | Adjusted odds ratio (95% CI) * | P value |
|---|---|---|---|---|---|---|---|
| Parent gender | Male | 73 (64) | 41 (36) | 1.23 (0.73 to 2.08) | 0.44 | 1.36 (0.76 to 2.44) | 0.30 |
| | Female | 94 (69) | 43 (31) | Reference | | Reference | |
| Parent age | 18–35 years | 53 (62) | 33 (38) | 2.49† (1.18 to 5.26) | 0.02 | 2.90† (1.30 to 6.44) | 0.01 |
| | 36–45 years | 62 (62) | 38 (38) | 2.45† (1.18 to 5.09) | 0.02 | 2.43† (1.12 to 5.28) | 0.03 |
| | ≥46 years | 52 (80) | 13 (20) | Reference | | Reference | |
| Parent living region | North of England | 46 (70) | 20 (30) | 0.80 (0.39 to 1.63) | 0.54 | 1.04 (0.49 to 2.22) | 0.92 |
| | Midlands England | 42 (65) | 23 (35) | 1.01 (0.50 to 2.03) | 0.98 | 1.03 (0.49 to 2.19) | 0.93 |
| | Wales, Scotland, Northern Ireland | 31 (76) | 15 (33) | 0.89 (0.41 to 1.95) | 0.78 | 1.15 (0.50 to 2.63) | 0.75 |
| | South of England | 48 (65) | 26 (35) | Reference | | Reference | |
| Parent household income | ≤ £34 999 | 68 (67) | 34 (33) | 0.95 (0.55 to 1.63) | 0.84 | 1.04 (0.55 to 1.95) | 0.91 |
| | ≥ £35 000 | 87 (65) | 46 (35) | Reference | | Reference | |
| Parent employment status‡ | Working | 131 (66) | 69 (35) | 1.26 (0.65 to 2.47) | 0.49 | 1.25 (0.56 to 2.77) | 0.59 |
| | Not working | 36 (71) | 15 (29) | Reference | | Reference | |
| Parent education level | ≤ A level | 88 (69) | 40 (31) | 0.82 (0.48 to 1.38) | 0.45 | 0.92 (0.51 to 1.64) | 0.77 |
| | ≥ Degree | 79 (64) | 44 (36) | Reference | | Reference | |
| My child has missed too much school since September this year§ | 5-point Likert scale (1=strongly agree, 5=strongly disagree) | n=166, M=3.11, SD=1.48 | n=83, M=3.17, SD=1.54 | 1.02 (0.86 to 1.22) | 0.79 | 1.03 (0.85 to 1.26) | 0.74 |
| My child is behind at school§ | 5-point Likert scale (1=strongly agree, 5=strongly disagree) | n=167, M=3.65, SD=1.35 | n=84, M=3.21, SD=1.57 | 0.81† (0.67 to 0.97) | 0.02 | 0.84 (0.69 to 1.02) | 0.08 |
| My child often says they have symptoms of illnesses when they do not§ | 5-point Likert scale (1=strongly agree, 5=strongly disagree) | n=167, M=3.89, SD=1.25 | n=84, M=3.54, SD=1.49 | 0.83† (0.68 to 0.99) | 0.05 | 0.84 (0.67 to 1.05) | 0.12 |
| Often no one is available to look after my child if they cannot go to school§ | 5-point Likert scale (1=strongly agree, 5=strongly disagree) | n=166, M=3.46, SD=1.44 | n=84, M=3.11, SD=1.55 | 0.85 (0.71 to 1.02) | 0.07 | 0.88 (0.72 to 1.07) | 0.20 |
| My child does not want to take time off school§ | 5-point Likert scale (1=strongly agree, 5=strongly disagree) | n=162, M=2.43, SD=1.17 | n=83, M=2.11, SD=1.14 | 0.78† (0.61 to 0.99) | 0.04 | 0.82 (0.63 to 1.05) | 0.12 |

Continued

**Table 1** Continued

| Parent demographics and attitudes about sending children to school while symptomatic | Level | Children were adherent (%) | Children were non-adherent (%) | Odds ratio (95% CI) | P value | Adjusted odds ratio (95% CI)* | P value |
|---|---|---|---|---|---|---|---|
| My child makes their own decisions about when they go to school§ | 5-point Likert scale (1=strongly agree, 5=strongly disagree)§ | n=166, M=4.25, SD=0.98 | n=83, M=3.60, SD=1.50 | **0.65¶ (0.52 to 0.81)** | <0.001 | **0.65¶ (0.50 to 0.83)** | <0.001 |
| My child should go to school if they have taken medication (eg, calpol, paracetamol)§ | 5-point Likert scale (1=strongly agree, 5=strongly disagree)§ | n=164, M=3.04, SD=1.30 | n=81, M=2.25, SD=1.20 | **0.60¶ (0.48 to 0.76)** | <0.001 | **0.57¶ (0.44 to 0.75)** | <0.001 |
| If children have common illnesses (eg, a cold), they should go to school§ | 5-point Likert scale (1=strongly agree, 5=strongly disagree)§ | n=166, M=2.30, SD=1.09 | n=83, M=2.13, SD=1.03 | 0.86 (0.66 to 1.11) | 0.24 | 0.83 (0.63 to 1.09) | 0.19 |
| Children build up their immunity by mixing with children who have common illnesses (eg, a cold)§ | 5-point Likert scale (1=strongly agree, 5=strongly disagree)§ | n=164, M=1.93, SD=0.88 | n=84, M=1.90, SD=1.03 | 0.97 (0.73 to 1.29) | 0.86 | 0.92 (0.67 to 1.26) | 0.61 |
| Other children with common illnesses (eg, a cold) go to school§ | 5-point Likert scale (1=strongly agree, 5=strongly disagree)§ | n=162, M=1.88, SD=0.79 | n=83, M=1.84, SD=0.99 | 0.96 (0.70 to 1.30) | 0.77 | 0.91 (0.65 to 1.27) | 0.58 |
| If children have mild symptoms of an illness, they should go to school§ | 5-point Likert scale (1=strongly agree, 5=strongly disagree)§ | n=162, M=1.88, SD=0.79 | n=80, M=2.46, SD=1.00 | **0.77† (0.59 to 0.99)** | 0.04 | **0.75† (0.56 to 0.99)** | 0.05 |
| Going to school is important for my child's mental health§ | 5-point Likert scale (1=strongly agree, 5=strongly disagree)§ | n=164, M=1.45, SD=0.67 | n=83, M=1.66, SD=0.93 | **1.41† (1.01 to 1.97)** | 0.04 | 1.43 (0.99 to 2.06) | 0.06 |
| When my child says they are too ill to attend school, I let them stay at home§ | 5-point Likert scale (1=strongly agree, 5=strongly disagree)§ | n=161, M=2.58, SD=0.67 | n=84, M=2.56, SD=1.15 | 0.98 (0.77 to 1.25) | 0.87 | 0.96 (0.74 to 1.25) | 0.78 |

*OR adjusted by parent gender, age, living region, household income, employment status and education level.
†P≤0.05 and formatted bold.
‡Working includes students and volunteers.
§The exact wording used in the survey.
¶P≤0.001 and formatted bold.
CI, confidence interval; M, mean; n, number of observations; SD, standard deviation.

**Table 2** Children who engaged in each activity reported on when they had stay-at-home symptoms (n=251)

| Exact statement wording, in the order that they were asked to participants | Children engaged in activity ('non-adherent'), n (%) | Children did not engage in activity ('adherent'), n (%) |
|---|---|---|
| Went to school | 60 (24%) | 191 (76%) |
| Went to a club or lesson outside of school | 29 (12%) | 222 (88%) |
| Visited someone from another household | 27 (11%) | 224 (89%) |
| Someone from another household visited the child | 15 (6%) | 236 (94%) |
| Someone from another household visited our household | 24 (10%) | 227 (90%) |

symptom, 33% (n=84, 95% CI: 28% to 39%) were non-adherent.

Most parents included in this study were women (55%, n=137), aged between 36 and 45 years (40%, n=100) and earned over £35 000 (57%, n=133) (males: 45%, n=114; aged 18–35 years: 34%, n=86, aged over 46 years: 26%, n=65; earned under £34 999: 43%, n=102). Table 1 presents associations between predictor variables and whether children went to school, engaged in other activities or socialised with non-household members (non-adherent). Non-adherence was moderately associated with younger parent age, with the odds of a child being non-adherent 2.9 times higher if the parent was aged 18–35 and 2.4 times higher if the parent was aged 36–45 compared with parents aged 46 or over.

Non-adherence was associated with parents' agreement with the following statements: their children made their own decisions about when to go to school; children should go to school if they take medication; and if children have a mild symptom of illness, they should go to school. Table 1 reports the exact statement wording, with lower values reflecting agreement with that statement.

Table 2 shows the frequencies of children's adherence and non-adherence for each of the behaviours that were then grouped into our outcome variable.

When rerunning the analyses to look at associations with school attendance only (sensitivity analysis), 227 parents were included. Fewer participants were included in the sensitivity analysis because not all 251 children attended school while symptomatic with stay-at-home symptoms. We found that school attendance was not associated with parent age, but it was associated with parents agreeing that their child did not want to take time off school. No other differences in associations were found between school attendance and non-adherence and our predictor variables (data not shown).

## DISCUSSION

We found that 33% of children were non-adherent to UK Government guidance, such as attending school or clubs or socialising with others when experiencing symptoms where it is advised to stay at home. These findings align with previous reports suggesting that this behaviour known as presenteeism, is common.[12–14] We also found that non-adherence was more likely for children with younger parents and for specific parental attitudes about symptoms of illness and their children's behaviour. This is concerning because children who attend school and socialise while unwell risk spreading their illness to others, increasing the likelihood that other children will need to take time off school and miss out on extracurricular and social activities, and putting vulnerable people that they come into contact with at risk.

We found that children were significantly more likely to be non-adherent when their parents believed that children should go to school when their symptoms were mild. This finding mirrors Woodland et al[12] and Webster et al[15] who found people were more likely to attend school or work with an infectious illness when they perceived their symptoms to be mild. We also found that children were more likely to be non-adherent if a parent felt that attendance at school was appropriate if analgesic medication had been taken. These findings suggest that a key consideration is whether the child will cope with school, rather than whether they might spread illness to others. These findings align with previous research that suggested families were less likely to self-isolate with symptoms that may indicate COVID-19 when parents, (1) perceived the symptoms to be mild, (2) were unsure of the cause of symptoms and (3) perceived a reduced severity of symptoms after taking analgesic medication.[25] A greater focus on the importance of decreasing social mixing when symptomatic may help reduce the incidence and spread of infectious disease outbreaks in schools and to others in the community.

Children were significantly more likely to be non-adherent where their parent reported that the child was responsible for making their own decisions about when they go to school. This mirrors the suggestion that children are more important in decision-making than previously recognised.[12] Educating children, particularly teenagers who may have a greater say over their attendance, about the need to remain at home when ill may be an important part of any future strategy to reduce presenteeism. But this finding may need to be taken with caution; the data were collected in the autumn term when schools reopened following national school closures due to the pandemic. Reports show that children were keen to get back to school at this time, which may have impacted this result.[26] Still, educating parents on these issues is also important. We found that parents who were aged 45 years or younger were significantly more likely to have a child that was non-adherent. We can presume that these parents have younger children, and therefore we expect parents to have more control over their child's activities

compared with older children. While younger parents may rely on more support from outside the household with childcare, in contrast to previous findings.[15] We did not find a significant association between children who were non-adherent and whether parents had someone available to look after their child if they could not go to school.[12 15]

We were surprised that we did not find a significant association between children who were non-adherent and parents who felt their child had missed too much school and were behind in school. This was also replicated in our sensitivity analyses that focused on children who had attended school and excluded all other types of non-adherent behaviours. Webster et al[15] found that people who had missed too much work and were concerned about their workload were more likely to attend work with an infectious illness. Woodland et al[12] also found that high motivations about school and high school absence were found to increase the risk of school-based presenteeism. We cannot be sure why our results contrast with previous research, but this could be due to the differences between parents not attending work (eg, financial implications) and children not attending school (eg, educational impacts) and pandemic-related reasons (eg, increased access to online schooling and activities). That being said, the adjusted odd ratio is 0.84 (95% CI: 0.69 to 1.02), which follows a trend that we expected, it could be that a significant association was not found due to a small sample size.

We suspected that there may have been an association between parents with higher education and children's non-adherence. Research indicates that parents who are educated may place higher importance on their children's education.[27] However, this was not the case, and similar levels of non-adherence were found; 31% for children with parents with a-level or below compared with 36% of children with parents with a degree or above. We suggest that the number of parents who are highly educated may be higher than the target population, which may have had an impact on these findings.

We found no significant associations between gender of the parental participant and reports of a child's adherence. Unfortunately, as children's demographic data were not collected, we were unable to explore associations with this variable in the study or other child demographic characteristics that may have an impact on our findings. For example, it has been suggested that children who are in transition years are at increased risk of school-based presenteeism.[12] Children need to be included in future studies about school-based presenteeism, as it is common for only the parents to be the participants.[12]

There were several limitations to this study. On average, 69% of children report at least one episode of school-based presenteeism,[12] which is higher than our finding of 33%. However, this average was taken from three studies, two of which measured presenteeism over 12 months whereas ours was assessed for less than 4 months. In addition, we asked parents to report about their child's most severe episode of recent illness and we only reported non-adherence for one child from each household for the 4 months that we measured. As such, we may have underestimated the prevalence of non-adherence: children who engaged in multiple bouts of non-adherence and households in which multiple children engaged in non-adherence will not have been identified. We also had a relatively small sample size and therefore we may not have detected small effects.[28] Our findings not only report on presenteeism behaviour but also interacting with others while symptomatic, therefore caution must be taken when comparing our findings to studies solely reporting on school-based presenteeism.

In addition, the data were drawn from a non-probability sample. Whether the sample was behaviourally and psychologically representative of the wider population of parents in the UK is unknown,[29] although we have no reason to suspect that the associations within the data cannot be generalised.[30] Our participants were mainly of white ethnicity (91%, n=224/246, n=5 were missing data) and we were unable to include ethnicity in our analysis due to small case numbers. Although this is fairly representative of the UK population, we were unable to compare results across different ethnic groups. We did not include child demographics in our analysis due to survey space limitations, thus we were unable to identify how children's characteristics may impact non-adherence. Our data collection occurred during a pandemic, with a substantial focus on the importance of remaining home when potentially infectious and after an extended period of home-schooling. How rates and predictors of presenteeism will change as we emerge from the pandemic is unknown.

## Conclusion

One-third of children with symptoms that indicated an infectious illness engaged in activities outside the home, including going to school or socialising with others. This behaviour goes against UK Government advice that is in place to prevent the spread of disease. We found that younger parents, children who made their own decisions about school attendance, children who had taken analgesic medication and having mild symptoms were risk factors for non-adherence. We suggest that parents and teenagers may benefit from guidance to help them make informed decisions about school attendance while unwell to reduce school presenteeism. Given, our study had a small sample size and was conducted during a pandemic, it is recommended that further research is needed to validate our findings in non-pandemic times and to identify possible interventions that may be generalisable to general circulating and seasonal infections.

**Acknowledgements** We would like to thank Dr Siobhan McAndrew, Honorary Senior Research Fellow, School of Sociology, Politics and International Studies, University of Bristol, for the dissemination of this study and assistance with the survey design.

**Contributors** All authors contributed to the conceptualisation of the study and approved the final draft. LW designed the survey, analysed the data and drafted the manuscript. LES, RA and JGR designed the survey and edited the manuscript. RW edited the manuscript. RW, RA and JGR supervised LW. JGR acted as the guarantor.

**Funding** This study was funded by the Economic and Social Research Council (grant number ES/P000703/1) and by the National Institute for Health and Care Research Health Protection Research Unit (NIHR HPRU) in Emergency Preparedness and Response (grant number NIHR200890), a partnership between the UK Health Security Agency, King's College London and the University of East Anglia. The views expressed are those of the author(s) and not necessarily those of the NIHR, UKHSA or the Department of Health and Social Care. For the purpose of open access, the author has applied a Creative Commons Attribution (CC BY) licence to any Author Accepted Manuscript version arising. The funders had no role in study design, data collection, data analysis, data interpretation or writing of the manuscript. The corresponding author had full access to all the data and had final responsibility for the decision to submit for publication.

**Competing interests** JGR, RA and LES participate in the UK's Scientific Advisory Group for Emergencies, or its subgroups. These groups did not fund the study or the authors. RA is an employee of the UK Health Security Agency. LW and RW have no competing interests to declare.

**Patient and public involvement** Patients and/or the public were not involved in the design, or conduct, or reporting, or dissemination plans of this research.

**Patient consent for publication** Not applicable.

**Ethics approval** This study involves human participants. The research was approved by the Faculty of Social Sciences and Law, University of Bristol (approval code 116976 and 8273). Participants gave informed consent to participate in the study before taking part.

**Provenance and peer review** Not commissioned; externally peer reviewed.

**Data availability statement** All data relevant to the study are included in the article or uploaded as online supplemental information. Our data is deposited in the King's College London research data repository, KORDS, (https://doi.org/10.18742/19519444). The data include sensitive information and cannot be made openly available. Requests for access to the data will be considered if made by academic teams willing to sign a standard access agreement and should be addressed to the King's College London research data repository team at research.data@kcl.ac.uk.

**ORCID iDs**
Lisa Woodland http://orcid.org/0000-0003-2440-3210
Louise E Smith http://orcid.org/0000-0002-1277-2564
Rebecca K Webster http://orcid.org/0000-0002-5136-1098

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
