## [Reviewer comments · BMJ Open]

ARTICLE DETAILS

TITLE (PROVISIONAL)	Why do children attend school, engage in other activities, or socialise when they have symptoms of an infectious illness? A cross-sectional survey
AUTHORS	Woodland, Lisa; Smith, Louise; Webster, Rebecca; Amlôt, Richard; Rubin, G James

VERSION 1 – REVIEW

REVIEWER	Most, Zachary M Children's Health System of Texas
REVIEW RETURNED	06-Mar-2023

GENERAL COMMENTS	Woodland and co-authors present a manuscript investigating what proportion of children attended school when they had symptoms of a contagious viral infection in the UK in Fall 2021, and whether there are any demographic or attitude based risk factors associated with this "risky behaviour". The authors utilized a cross-sectional survey to evaluate this association. The authors found that out 251 children with at least one such symptom, 84 (33%) had attended school while sick those symptoms. They found that this "risky behaviour" was associated with having younger parents, if the children made their own decisions to attend school or not, if the parents thought that children who take over the counter cold remedies should go to school, and if the parents thought that children with mild symptoms should go to school. The authors conclude that school attendance among symptomatic children that goes against the guidance of the UK government is common and interventions to reduce this could be targeted at education around school presenteeism. Overall, this is an interesting topic of research for which there are little data available. Clearly, such presenteeism and attitudes regarding presenteeism will vary greatly in different geographic areas so gathering such data to help guide regional public health interventions is important. The biggest limitation of this study is in its methodology. The survey was conducted using a convenience sample which likely introduces bias into the results so it remains unclear how accurate it would be to apply these results to the UK population as a whole. That does not exclude this from being a useful study and this could still be an important contribution to the literature on school presenteeism. However, there are several issues with the manuscript that should be addressed prior to publication. I have labeled the following lists 'Major Comments' and 'Minor Comments'. 'Major Comments' refer to issues that I think need to be addressed prior to publication of the manuscript. 'Minor Comments' refer to suggestions that I have to improve the manuscript but are not crucial to its acceptance for publication.
--

	I hope that you find these comments helpful and keep up the strong work. Note that the page number of all comments reference the page number of the proof, not the page number generated by the authors Major Comments  • Title • Abstract  o In the participants section of the abstract, the authors should describe the study population (e.g. Eligible participants were between 18 and 75-years old, lived in the UK, and had at least one child between 4 and 17 years-old at home who had a “stay-at-home” symptom since September 2019. Participants were recruited from a pre-existing pool of potential respondents who had already expressed an interest in receiving market research surveys”). What is currently written would fit better in outcome measures o In addition, the outcome measures section should mention that in addition to “risky behavior”, demographic and attitude factors were measured o As in the main section of the text (see my comments for the methods section later) please include the number of surveys that were sent out along with the response rate. • Introduction  o The last paragraph of the introduction should more explicitly mention the aim or objective of this study, rather than just listing the methods. Try to answer the question of: Why did you perform this study? Why is it important and what does it add to the current literature? For example, “In this study, we sought to determine what factors contribute to the decision for children to attend school or other social functions when sick with an infectious illness.” • Methods  o Page 5 Line 60 – Page 6 Line 16: Please give a justification for why non-probability sampling was used and please describe the sampling method in more detail. How were these sampling quotas determined? It seems that you combined age with 4 different demographics to define these strata, which would mean you had, at a minimum, 8 different strata with quotas (but potentially many more than that if there were more than 2 age groups, government office regions, working statuses, and social grades). With only 4,962 participants, it seems unlikely that you could have adequately filled that many strata. Please provide more information on the size of the sampling quotas for each strata and whether the intended sizes were actually met with the number of survey respondents. How did participants end up on this “pre-existing pool of potential respondents”? The use of a convenience sample would likely bias the results and should be listed as a weakness o It may be helpful for you to explicitly define what was your target population (who you wanted to learn about) and your study population (who you actually sampled/measured). o Page 6 Line 18-20: I see that 4,962 participants completed the survey, but it is crucial to also know how many people had the survey sent to them (i.e., what was the survey response rate?). It would affect how we can understand this study if 50% of those sent the survey responded, vs 5%. • Results  o Page 8 Line 24: Please consider adding data on demographics for the parents (or households) who responded to the survey. This
--	--

	is not as ideal as having demographics on the children, but at least it can put this study into some context to evaluate its external validity. You have these data in table 1, but the text should include a brief summary of the demographics of your sample. o Page 8 Line 31-35: In addition to saying that risky behaviour was associated with those factors, please describe the direction of the association (positive or negative correlation?) and consider adding a qualifying term to describe how large the association was (e.g. strongly associated). The data are in the table, but the text of the manuscript should be interpretable even without looking at the table. o Please provide results on how many of the included children partook in which risky behaviour (attending school, attending clubs, attending other households, having visitors in the home, etc) • Discussion o Page 12 Line 16-21: Thank you for discussing how non-probability sampling is a limitation of this study. I disagree with the statement that there is no reason to suspect that the associations cannot be generalised. It is unclear from the data whether the sample reflects similar demographics to the UK as a whole. Also, your sample is clearly biased towards the type of people who were reached with advertising described in the methods. Additionally, in a voluntary survey there will almost always be a bias towards more health-motivated people participating in the survey (healthy volunteer bias). It is possible that the people who did not participate in the survey engage in even more 'risky behaviours'. o Page 12 Line 6-30: For each of the limitations please discuss how these limitations affect the quality of this study. For example, by having 91% of respondents be of white ethnicity, is the problem that the sample was not well representative of the UK population (a quick search found that the UK population is ~88% white, so this is actually fairly representative) or is it that you were not able to evaluate whether the association you found was robust across different racial/ethnic groups (which is more of a problem of equity and representation in epidemiologic studies than with external validity). This is just one example. Please clarify why these are limitations for your study and what impact they should have interpreting the results. o Page 12 Line 39-41: The final sentence in the conclusion says that parents and teenagers need guidance... I think that need is too strong of a word for this sentence based on the study results. This is an important study, but it has substantial limitations on how accurate these data are and with external validity. What we really need is more studies with a more rigorous sampling procedure to better understand the true rate of risky behaviour. Additionally, your study does not provide any evidence that "guidance" to parents and teenagers would affect their behaviour. Minor Comments • Title • Abstract o In the first sentence of the results section the "stay-at-home" symptoms are defined "either recent vomiting, diarrhoea, high/temperature...". This description should be in the "Outcome measures" section as it is describing a key definition and is really part of the methods and not results o The definition of "engaged in risky behaviour" should be more explicit. Based on the way it is written, it could be interpreted that attending school or socialising while symptomatic was just one of
--	---

	many types of “risky behaviour” evaluated, but in fact those were the only “risky behaviours” evaluated  • Introduction  o Page 5 Line 18-21: The comment that children are susceptible to infections because “their immune systems are immature” should either be edited, removed, or better substantiated by referenced evidence. While infants may have “immature” immune systems, school-aged children usually have quite robust immune systems. School-aged children often do not have a wide array of memory T and B cells which may make them more susceptible to viral infections, but this is not what I would think when I hear the term “immature” immune system. o Page 5 Line 18-21: In the same sentence, I would not say that children “have a poor understanding of hygiene practices”. Rather, young children engaging in developmentally normal behaviors (such as attending congregate settings [school], sharing toys, touching their faces, and for infants/toddlers putting objects in their mouths) may put them at risk for viral infections because such behaviors would not meet the standards of “good hygiene” for older children and adults. In fact, I think that many school-aged older children and adolescents actually have a better understanding of hygiene practices than adults do, but they are still at higher risk for viral infections because they must attend school with close contact with many other individuals. o Page 5 Line 27-28: The term “worrying proportion” is subjective and more objective terminology would be better (e.g. “a large proportion”) o Page 5 Line 33-39: Citation number 11 is a systematic review that covers a similar topic to your study, Since your study is smaller and also observational, you should provide some justification in the introduction as to why your study was important or needed despite the systematic review already being published (i.e. please describe what is new or different about your study to justify to the reader why they should read this study instead of the systemic review that you site) • Methods  o Page 5 Line 53: “a wider piece of work” is vague and perhaps a short but more descriptive term could be used here o Why was there a quota stratum for gender? The gender of the respondent does not reflect the gender of the child or whether there is another parent in the home of the same or another gender. It is unclear why we might think the gender of the respondent would affect these results. If you do think it may have an effect, it would likely only be on the accuracy of the data (e.g. recall bias could potentially be different between genders). However, later in the discussion you do not discuss any concerns with the possibility of information bias/inaccurate data. o Page 6 Line 18-20: these sentences might belong better in the results section, along with a figure showing a diagram starting from how many surveys were sent out to your final sample of 251 households o Page 6 Line 20: please clarify the line “experienced symptom(s) that required them to stay at home”. By this do you mean that they had symptoms that, according to recommendations from the UK government, they should have stayed home? Clearly not all of these respondents actually kept their children home for these events o Page 6 Line 35-36: The lack of demographic data collected on the children in this survey is a major weakness of this study and
--	--

	makes it difficult to interpret. This should be addressed in the discussion  o Page 6 Line 48-50: The use of only the most severe episode may have affected the results (for instance, parents may have sent their children to school more frequently for milder illnesses, so your final estimate of how often sick children attended school may be an underestimate). This should be addressed in the discussion. o Page 7 Line 32-33: Typically, the reference to Table 1, as you make here, would be done in the results section and not the methods section o Page 7 Line 40-42: This was mentioned earlier and is redundant here o I find the term “risky behaviour” to be a bit strong here. To me it implies that the children are choosing to take risks, however without data on the age of the children that is hard to say. Certainly a 7-year-old is not really deciding for themselves whether to go to school or not. In that case it is really the parents who are choosing the risk. It may be better to just stick with a term that describes exactly what these children did (e.g. attended school or socialising with others)  • Results  o Page 9: Is the ‘Parent gender’ the gender of whichever parent answered the survey? Does it imply anything about whether there were one or two adults in the household and which adult was the primary caretaker? o In Table 1 for the Likert scale values the table lists N, M, and SD in each cell. I assume that is Number of observations, mean, and standard deviation. Please clarify this in the footnotes, or add some sort of description in each row of the first column o For the Likert scale results, it is a little confusing at first glance because the lower values on the Likert scale indicate more agreement with the statement. So, odds ratios less than one correlate with more risk behavior with more agreement to that attitude. It was a little confusing at first because I would normally expect higher odds ratios (greater than 1) to show a direct correlation between the attitude and high-risk behavior. It is ok as it is now, but please consider adding text to the results section to clarify this. o Putting a star next to the P value and bolding the text for P values less than 0.05 is a bit redundant. The stars don’t actually add much value to this table • Discussion  o Can you put your result of 33% attending school or a social function while sick in context? Are there any external data to compare this to? You describe this as being alarmingly high, but frankly I am surprised that it is not higher (I suspect the true rate is higher than this). Maybe no other studies are available on this topic to compare it to, but if you could find a study that would help a lot to put your results into context.
--	--

REVIEWER	Li, Dan Yale University School of Public Health
REVIEW RETURNED	19-May-2023

GENERAL COMMENTS	Why Do Children Attend School and Socialise When They Have Symptoms Of An Infectious Illness? A Cross-Sectional Survey The manuscript aims to explore the prevalence, risk factors, and implications of children attending school and engaging in social activities while exhibiting symptoms of an infectious illness,
---

	emphasizing the necessity for improved guidance to parents and children regarding recognizing symptoms and taking appropriate measures to prevent infection transmission. INTRODUCTION: Minor need for improvement: Objectives: While the stated objectives are clear and specific, it would be beneficial to include any prespecified hypotheses, if applicable. Including hypotheses can provide additional clarity and guidance to readers regarding the study's research questions. METHODS: The manuscript might flow better if it were structured as follows: Variables: Minor improvement needed. The report should provide more specific details about the definitions of outcomes, exposures, predictors, and potential confounders. Including the diagnostic criteria for the 14 symptoms of infectious illnesses would also enhance the clarity of the study. Data sources/measurement: Minor improvement needed. The report should provide more specific details about the methods of assessment and explain the comparability of assessment methods if there are multiple groups. Bias: Major improvement needed. The report should explicitly describe efforts made to address potential sources of bias. This is an important aspect of study methodology that should be addressed. Study size: Major improvement needed. The report should explain how the study size was determined. Quantitative variables and statistical methods: Major improvement needed. The report should provide more specific details about the statistical methods used to control for confounding, examine subgroups and interactions, handle missing data, and account for the sampling strategy. RESULTS: The manuscript might flow better if it were structured as follows: Participants: (a) There is a need to report the numbers of individuals at each stage of the study, including eligibility, inclusion, follow-up, and analysis stages. This is a major need for improvement as it provides transparency and allows readers to understand the sample size and potential biases. (b) The reasons for non-participation at each stage should be provided. This is a minor need for improvement as it helps identify potential sources of bias or limitations in the study. Descriptive data: (a) The characteristics of study participants, such as demographic, clinical, and social information, should be reported. (b) The number of participants with missing data for each variable of interest should be indicated. Main results: (a) Unadjusted estimates are provided, but the report should also include confounder-adjusted estimates and their precision. This is a major need for improvement as it allows for a more robust analysis and interpretation of the results.
--	---

	(b) Category boundaries should be reported when continuous variables are categorized. However, since no continuous variables are mentioned in the given information, this is not applicable. (c) The translation of estimates into absolute risk for a meaningful time period should be considered if relevant. It is not mentioned in the given information. Other analyses: The sensitivity analysis is reported, which is a positive aspect of the study.
--	---

REVIEWER	Cowger , Tori L Harvard T.H. Chan School of Public Health
REVIEW RETURNED	10-Jun-2023

GENERAL COMMENTS	Thank you for the opportunity to review your manuscript. I have outlined my comments and recommendations below, including major comments that I believe would need to be addressed before publication and minor comments that might improve clarity and strengthen this work. MAJOR COMMENTS: 1. Sampling frame, selection, and target population The authors report that their sample was selected via an internet survey among a non-probability sample of 5,000 participants recruited from a pool of participants interested in completing market research surveys and through recruitment via advertising on various online platforms. Authors also state that this survey was conducted as a “wider piece of work” (page 5 of 23, line 52), however, they do not state the objective or aims of primary study for which this sample was recruited. Authors describe quotas based on strata of demographic factors including age, working status, and social grade, however, they do not state whether the population sampled met these recruitment targets nor do they present any descriptive statistics of the full study population compared to their target population. This information is essential in interpreting the results of this survey, especially since it seems likely that the population of individuals who are interested and available to complete market research surveys is likely to differ in key ways from the general population (e.g., access to resources, health information, education, etc.) that are likely to impact whether or not a parent sends a child to school while symptomatic. In addition, authors state that of their 4,962 participants, 941 had at least one school-aged child, with 251 of these participants reporting that their child had stay-at-home symptoms and were therefore included in the primary analysis. This is a relatively small subset of the survey population, and it would be helpful to know how the population included in the analysis compares to both the target population and other subsets of the study population (e.g., parents who had children that did not report symptoms). On page 12 of 23, authors acknowledge non-probability sampling as a limitation, however they go on to say that they “have no reason to suspect that the associations within the data cannot be generalized.” I have two key concerns related to this statement, as a highly selected sample presents threats to both internal and external validity.
---

First, the authors acknowledge some differences between their study population and their target population and it seems like the final sample may not be representative of the general population – For example, e.g., nearly half $n=123/151$ of parents of symptomatic children have a degree or higher education levels, which is substantially higher than reported in UK census data (~33%). In addition, authors report that they could not report results by race/ethnicity as more than 90% of the study population were of white ethnicity. The over-representation of certain groups in the study sample has the potential to bias the estimate of prevalence of school attendance while symptomatic if these characteristics associated with selection into the sample are associated with symptomatic school attendance. In addition, differences between sample and target population can represent threats to internal validity and bias estimates of association if sample selection is associated with both the exposures of interest in the study (i.e., characteristics presented in table 1), and their outcome, symptomatic school attendance. It seems likely that characteristics associated with selection into the sample (e.g., parental education) would be associated with both perceptions of illness and also associated to sending children to school when sick. It would be helpful to understand the magnitude of this potential bias by comparing the characteristics of the study population those available in the target population. I would suggest that where possible, authors provide a table comparing the sociodemographics of their study population with that of their target population where publicly available data are available. In addition, it would be helpful to know how the final sample in this analysis compares to the full population sample and relevant subsets. To address this, authors could present a supplementary table comparing the demographic characteristics they currently present in table 1 for the full study population (~N=5000), the population of all parents in the full sample (n=941), and parents of children with symptoms to get a better sense of how similar or different these populations are. This may help give a better sense of the magnitude of potential bias that might be introduced during study recruitment and through exclusions.

Second, the authors mention the “wider population of parents in the UK” which is presumably their target population, and seem to be discussing the internal validity of their results, but use the term “generalize” which is more appropriately applied when discussing external validity (e.g., populations of parents outside of the UK). Authors should consider being more precise in their language.

2. Framing and language of “Engaging in risky behavior”
Throughout their paper, the authors refer to children who attend school or activities outside of the household as “engaging in risky behavior.” This framing and language seems vague and also has the potential to stigmatize children and their parents. I would suggest authors consider using different language that is more specific to the outcome the authors measured, for example, “attending school while symptomatic,” “socializing while symptomatic” or “school-based presenteeism,” which authors refer to in their introduction.

3. Missing results of sensitivity analyses.
Authors mention a sensitivity analysis wherein they considered the outcome of school attendance only (vs. their primary analysis which considered school attendance together with attending other

activities), however, the results of these analysis are not presented in detail anywhere. On page 8 of 23, the authors state that n=227 parents were included in this analysis, but do not explain why their sample size is different for the sensitivity analysis vs. their main analysis. The authors also note several key differences between primary and sensitivity analyses, but without seeing the full results of the sensitivity analyses presented, these results are hard to interpret. Authors should consider presenting the results of the sensitivity analysis in full in their supplementary material and should explain why the sample sizes are different between primary and sensitivity analyses.

Minor Comments:

1. Descriptive statistics of stay-at-home symptoms: Authors list out 14 symptoms included in the survey on page 19 of 23 and list out several as “stay at home” symptoms per UK guidelines on page 7 of 23. On page 8 of their results section, authors report that 48% of parents (n=454/941) reported their child reported at least one symptom, but only 27% (n=251) reported a stay-at-home symptom. This seems like a big difference to me, and it may be helpful to include a supplemental table that delineates which of the symptoms surveyed were considered ‘stay-at-home’ symptoms per study criteria and which symptoms were most common among study participants reporting symptoms. This would help readers better understand the study population included in the sample and symptoms experienced by those included and excluded from the study.

2. Description of symptom/symptoms: it would be helpful to know which and how many symptoms children included in the study were experiencing and whether that was related to school attendance while symptomatic. For example, are children with certain sets of symptoms (e.g., cough only) more likely to be sent to school than children with different symptoms? I suggest that the authors add both stay-at-home symptom type and number of stay-at-home symptoms to Table 1.

3. Power and statistical significance: the final sample size included in the study is relatively small, which likely impacts the statistical power to detect differences between groups in the study. However, authors do not mention this as a limitation, and there are several instances where authors interpret lack of statistical significance as an absence of a true association rather than lack of statistical power. For example, on page 12 of 23, authors describe a null association between perceptions of being behind in school and attending school while symptomatic and interpret this as contrasting with prior literature. However, this relationship was statistically significant in univariate analyses, with similar point estimates in both unadjusted and adjusted analyses. Therefore, this seems likely an issue of statistical power rather than truly conflicting with prior research, and authors should consider modifying language to reflect this.

4. Multiple testing: The authors report results of many statistical tests in their primary analyses and presumably conducted additional statistical tests in their supplementary analyses. However, I didn’t see if the authors mentioned their approach or rationale for multiple testing?

VERSION 1 – AUTHOR RESPONSE

Reviewer: 1

Dr. Zachary M. Most, Children's Health System of Texas

Comments to the Author:

Woodland and co-authors present a manuscript investigating what proportion of children attended school when they had symptoms of a contagious viral infection in the UK in Fall 2021, and whether there are any demographic or attitude based risk factors associated with this "risky behaviour". The authors utilized a cross-sectional survey to evaluate this association. The authors found that out of 251 children with at least one such symptom, 84 (33%) had attended school while sick those symptoms. They found that this "risky behaviour" was associated with having younger parents, if the children made their own decisions to attend school or not, if the parents thought that children who take over the counter cold remedies should go to school, and if the parents thought that children with mild symptoms should go to school. The authors conclude that school attendance among symptomatic children that goes against the guidance of the UK government is common and interventions to reduce this could be targeted at education around school presenteeism.

Overall, this is an interesting topic of research for which there are little data available. Clearly, such presenteeism and attitudes regarding presenteeism will vary greatly in different geographic areas so gathering such data to help guide regional public health interventions is important. The biggest limitation of this study is in its methodology. The survey was conducted using a convenience sample which likely introduces bias into the results so it remains unclear how accurate it would be to apply these results to the UK population as a whole. That does not exclude this from being a useful study and this could still be an important contribution to the literature on school presenteeism. However, there are several issues with the manuscript that should be addressed prior to publication.

I have labelled the following lists 'Major Comments' and 'Minor Comments'. 'Major Comments' refer to issues that I think need to be addressed prior to publication of the manuscript. 'Minor Comments' refer to suggestions that I have to improve the manuscript but are not crucial to its acceptance for publication.

I hope that you find these comments helpful and keep up the strong work.

Note that the page number of all comments reference the page number of the proof, not the page number generated by the authors.

Major Comments

- Title
- Abstract

o In the participants section of the abstract, the authors should describe the study population (e.g. Eligible participants were between 18 and 75-years old, lived in the UK, and had at least one child between 4 and 17 years-old at home who had a "stay-at-home" symptom since September 2019. Participants were recruited from a pre-existing pool of potential respondents who had already expressed an interest in receiving market research surveys"). What is currently written would fit better in outcome measures

Response: We have updated the abstract as you have suggested.

o In addition, the outcome measures section should mention that in addition to "risky behavior", demographic and attitude factors were measured

Response: We have added to the abstract that demographic and attitude factors were also measured.

o As in the main section of the text (see my comments for the methods section later) please include the number of surveys that were sent out along with the response rate.

Response: Recruitment followed a complex pattern, in which responders from earlier waves were invited to participate and additional respondents from the panel were then invited to take part to make up for non-respondents. Importantly, respondents were drawn from a panel that was originally self-selected. Providing a response rate therefore gives no information as to the level of non-response bias present in our sample as compared to the general population. In addition, quota samples aim to minimise response bias by filling predetermined targets so that the social and personal characteristics of the participants match those of the national population. As such, participants who belong to a quota that has already been met are prevented from completing the survey. Therefore, response rates are not useful indicators of response bias in quota samples and are not usually reported. We have updated the manuscript to include this rationale.

- Introduction

o The last paragraph of the introduction should more explicitly mention the aim or objective of this study, rather than just listing the methods. Try to answer the question of: Why did you perform this study? Why is it important and what does it add to the current literature? For example, “In this study, we sought to determine what factors contribute to the decision for children to attend school or other social functions when sick with an infectious illness.”

Response: We have updated the last paragraph of the introduction as you have suggested.

- Methods

o Page 5 Line 60 – Page 6 Line 16: Please give a justification for why non-probability sampling was used and please describe the sampling method in more detail. How were these sampling quotas determined? It seems that you combined age with 4 different demographics to define these strata, which would mean you had, at a minimum, 8 different strata with quotas (but potentially many more than that if there were more than 2 age groups, government office regions, working statuses, and social grades). With only 4,962 participants, it seems unlikely that you could have adequately filled that many strata. Please provide more information on the size of the sampling quotas for each strata and whether the intended sizes were actually met with the number of survey respondents.

How did participants end up on this “pre-existing pool of potential respondents”? The use of a convenience sample would likely bias the results and should be listed as a weakness.

Response: We have edited the relevant text to make it clear that only age and gender were interlocked. We have provided more information about the sample. We have also provided a reference that describes: the survey company’s profile; sample source and recruitment; sampling and project management; data quality and validation; policies and compliance; and metrics. Participants were recruited from the UK panel, we were unable to gather the data for the 2020/2021 panel size, although the 2023 panel comprises of $n = 345,130$ people.

o It may be helpful for you to explicitly define what was your target population (who you wanted to learn about) and your study population (who you actually sampled/measured).

Response: We have added this to the methods.

o Page 6 Line 18-20: I see that 4,962 participants completed the survey, but it is crucial to also know how many people had the survey sent to them (i.e., what was the survey response rate?). It would affect how we can understand this study if 50% of those sent the survey responded, vs 5%.

Response: Please see the above response that describes the rationale for not providing this data.

- Results

o Page 8 Line 24: Please consider adding data on demographics for the parents (or households) who responded to the survey. This is not as ideal as having demographics on the children, but at least it can put this study into some context to evaluate its external validity. You have these data in table 1, but the text should include a brief summary of the demographics of your sample.

Response: We have added parent gender, age and household income to provide study context.

o Page 8 Line 31-35: In addition to saying that risky behaviour was associated with those factors, please describe the direction of the association (positive or negative correlation?) and consider adding a qualifying term to describe how large the association was (e.g. strongly associated). The data are in the table, but the text of the manuscript should be interpretable even without looking at the table.

Response: We have added this information as you have suggested.

o Please provide results on how many of the included children partook in which risky behaviour (attending school, attending clubs, attending other households, having visitors in the home, etc)

Response: We have added a results table that includes the activities that children engaged in while symptomatic.

• Discussion

o Page 12 Line 16-21: Thank you for discussing how non-probability sampling is a limitation of this study. I disagree with the statement that there is no reason to suspect that the associations cannot be generalised. It is unclear from the data whether the sample reflects similar demographics to the UK as a whole. Also, your sample is clearly biased towards the type of people who were reached with advertising described in the methods. Additionally, in a voluntary survey there will almost always be a bias towards more health-motivated people participating in the survey (healthy volunteer bias). It is possible that the people who did not participate in the survey engage in even more 'risky behaviours'.

Response: We agree that it is possible that the absolute percentages of behaviour that we report were affected by bias. However, the question here is different. It is whether the associations that we found between our predictor variables and non-adherence are affected by bias. The paper that we have cited discusses this in some detail. We would need to assume that the association between presenteeism and e.g. believing that your child is behind at school is stronger or weaker in people who volunteer for a study and people who do not. This is possible, of course, but we suspect it is less likely.

o Page 12 Line 6-30: For each of the limitations please discuss how these limitations affect the quality of this study. For example, by having 91% of respondents be of white ethnicity, is the problem that the sample was not well representative of the UK population (a quick search found that the UK population is ~88% white, so this is actually fairly representative) or is it that you were not able to evaluate whether the association you found was robust across different racial/ethnic groups (which is more of a problem of equity and representation in epidemiologic studies than with external validity). This is just one example. Please clarify why these are limitations for your study and what impact they should have interpreting the results.

Response: We have added more detail about how the limitations that we have reported about may impact our findings.

o Page 12 Line 39-41: The final sentence in the conclusion says that parents and teenagers need guidance... I think that need is too strong of a word for this sentence based on the study results. This is an important study, but it has substantial limitations on how accurate these data are and with external validity. What we really need is more studies with a more rigorous sampling procedure to

better understand the true rate of risky behaviour. Additionally, your study does not provide any evidence that “guidance” to parents and teenagers would affect their behaviour.

Response: We have reduced the strength of the statement by adding “we suggest that.” We have also added a sentence reporting the main limitations of our findings and that further research is needed to validate our findings and to identify possible interventions.

Minor Comments

- Title
- Abstract
- o In the first sentence of the results section the “stay-at-home” symptoms are defined “either recent vomiting, diarrhoea, high/temperature...”. This is description should be in the “Outcome measures” section as it is describing a key definition and is really part of the methods and not results

Response: We have moved this definition to outcome measures.

- o The definition of “engaged in risky behaviour” should be more explicit. Based on the way it is written, it could be interpreted that attending school or socialising while symptomatic was just one of many types of “risky behaviour” evaluated, but in fact those were the only “risky behaviours” evaluated

Response: We have altered the wording to make the definition more explicit.

- Introduction
- o Page 5 Line 18-21: The comment that children are susceptible to infections because “their immune systems are immature” should either be edited, removed, or better substantiated by referenced evidence. While infants may have “immature” immune systems, school-aged children usually have quite robust immune systems. School-aged children often do not have a wide array of memory T and B cells which may make them more susceptible to viral infections, but this is not what I would think when I hear the term “immature” immune system.

Response: We have changed the wording to children’s immune systems are “developing” and added another reference.

- o Page 5 Line 18-21: In the same sentence, I would not say that children “have a poor understanding of hygiene practices”. Rather, young children engaging in developmentally normal behaviours (such as attending congregate settings [school], sharing toys, touching their faces, and for infants/toddlers putting objects in their mouths) may put them at risk for viral infections because such behaviours would not meet the standards of “good hygiene” for older children and adults. In fact, I think that many school-aged older children and adolescents actually have a better understanding of hygiene practices than adults do, but they are still at higher risk for viral infections because they must attend school with close contact with many other individuals.

Response: We have altered this wording to show that children are capable of understanding hygiene practices although are at risk for viral infections.

- o Page 5 Line 27-28: The term “worrying proportion” is subjective and more objective terminology would be better (e.g. “a large proportion”)

Response: We have changed the wording to a “large” proportion.

- o Page 5 Line 33-39: Citation number 11 is a systematic review that covers a similar topic to your study, Since your study is smaller and also observational, you should provide some justification in the introduction as to why your study was important or needed despite the systematic review already being published (i.e. please describe what is new or different about your study to justify to the reader why they should read this study instead of the systemic review that you site)

Response: We have added that the systematic reviews do not report significant risk factors associated with school-based presenteeism in UK children.

- Methods

- o Page 5 Line 53: “a wider piece of work” is vague and perhaps a short but more descriptive term could be used here

Response: We have provided more context about the study that this was a part of.

- o Why was there a quota stratum for gender? The gender of the respondent does not reflect the gender of the child or whether there is another parent in the home of the same or another gender. It is unclear why we might think the gender of the respondent would affect these results. If you do think it may have an effect, it would likely only be on the accuracy of the data (e.g. recall bias could potentially be different between genders). However, later in the discussion you do not discuss any concerns with the possibility of information bias/inaccurate data.

Response: Gender was a quota set across each wave of the longitudinal study and we are not sure if parent gender would have an association with non-adherence. However, after this study was conducted a systematic review that we also conducted suggested that parent gender may impact the findings. We have added the findings of this systematic review to the introduction (for clarity) and the possibility of gender bias to the discussion.

- o Page 6 Line 18-20: these sentences might belong better in the results section, along with a figure showing a diagram starting from how many surveys were sent out to your final sample of 251 households

Response: We have moved this paragraph to the results section.

- o Page 6 Line 20: please clarify the line “experienced symptom(s) that required them to stay at home”. By this do you mean that they had symptoms that, according to recommendations from the UK government, they should have stayed home? Clearly not all of these respondents actually kept their children home for these events

Response: We have added the definition of stay-at-home symptoms to the results section.

- o Page 6 Line 35-36: The lack of demographic data collected on the children in this survey is a major weakness of this study and makes it difficult to interpret. This should be addressed in the discussion.

Response: We have addressed this in the discussion.

Page 6 Line 48-50: The use of only the most severe episode may have affected the results (for instance, parents may have sent their children to school more frequently for milder illnesses, so your final estimate of how often sick children attended school may be an underestimate). This should be addressed in the discussion.

Response: We have added to the discussion that we may have underestimated the prevalence of non-adherence.

- o Page 7 Line 32-33: Typically, the reference to Table 1, as you make here, would be done in the results section and not the methods section

Response: We agree that the reference to Table 1(changed to Table 2) here is uncommon. However, as this was a short paper it felt repetitive to report the demographic information again, so we have directed the readers to the results table instead.

- o Page 7 Line 40-42: This was mentioned earlier and is redundant here

Response: We have removed page 7 line 40 to 42.

o I find the term “risky behaviour” to be a bit strong here. To me it implies that the children are choosing to take risks, however without data on the age of the children that is hard to say. Certainly a 7-year-old is not really deciding for themselves whether to go to school or not. In that case it is really the parents who are choosing the risk. It may be better to just stick with a term that describes exactly what these children did (e.g. attended school or socialising with others)

Response: We have relabelled our outcome variable to non-adherent behaviour as another peer reviewer suggested that this terminology could be stigmatising.

• Results

o Page 9: Is the ‘Parent gender’ the gender of whichever parent answered the survey? Does it imply anything about whether there were one or two adults in the household and which adult was the primary caretaker?

Response: We asked how many children are “currently living or staying at this address for at least two months,” it does not imply any further demographic information.

o In Table 1 for the Likert scale values the table lists N, M, and SD in each cell. I assume that is Number of observations, mean, and standard deviation. Please clarify this in the footnotes, or add some sort of description in each row of the first column

Response: We have added the definition of these abbreviations to the table.

o For the Likert scale results, it is a little confusing at first glance because the lower values on the Likert scale indicate more agreement with the statement. So, odds ratios less than one correlate with more risk behaviour with more agreement to that attitude. It was a little confusing at first because I would normally expect higher odds ratios (greater than 1) to show a direct correlation between the attitude and high-risk behaviour. It is ok as it is now, but please consider adding text to the results section to clarify this.

Response: We have added a statement to clarify that lower values reflect agreement to the statement.

o Putting a star next to the P value and bolding the text for P values less than 0.05 is a bit redundant. The stars don’t actually add much value to this table

Response: We have kept the asterisks to show the difference between P values of less than 0.05 and 0.0001.

• Discussion

o Can you put your result of 33% attending school or a social function while sick in context? Are there any external data to compare this to? You describe this as being alarmingly high, but frankly I am surprised that it is not higher (I suspect the true rate is higher than this). Maybe no other studies are available on this topic to compare it to, but if you could find a study that would help a lot to put your results into context.

Response: We have added a reference that found 69% of children reported at least one-episode of presenteeism and described this in relation to our study’s findings.

Reviewer: 2

Miss Dan Li, Yale University School of Public Health

Comments to the Author:

Why Do Children Attend School and Socialise When They Have Symptoms Of An Infectious Illness?
A Cross-Sectional Survey

The manuscript aims to explore the prevalence, risk factors, and implications of children attending school and engaging in social activities while exhibiting symptoms of an infectious illness, emphasizing the necessity for improved guidance to parents and children regarding recognizing

symptoms and taking appropriate measures to prevent infection transmission.

INTRODUCTION:

Minor need for improvement:

Objectives: While the stated objectives are clear and specific, it would be beneficial to include any prespecified hypotheses, if applicable. Including hypotheses can provide additional clarity and guidance to readers regarding the study's research questions.

Response: We have added a sentence that describes our hypothesis.

METHODS: The manuscript might flow better if it were structured as follows:

Variables: Minor improvement needed. The report should provide more specific details about the definitions of outcomes, exposures, predictors, and potential confounders. Including the diagnostic criteria for the 14 symptoms of infectious illnesses would also enhance the clarity of the study.

Response: We have added more details about the definitions of the variables. We have also clarified that the 14 symptoms of infectious illness were "listed by the UK Government" to show adherent and non-adherent behaviours rather than a diagnostic measure.

Data sources/measurement: Minor improvement needed. The report should provide more specific details about the methods of assessment and explain the comparability of assessment methods if there are multiple groups.

Response: The wording of all questions is already provided in full in the supplementary material. All participants received the same assessments.

Bias: Major improvement needed. The report should explicitly describe efforts made to address potential sources of bias. This is an important aspect of study methodology that should be addressed.

Response: We have reported the response bias and more information about how the participants were recruited for transparency.

Study size: Major improvement needed. The report should explain how the study size was determined.

Response: We have explained that we did not determine the sample size prior to the study.

Quantitative variables and statistical methods: Major improvement needed. The report should provide more specific details about the statistical methods used to control for confounding, examine subgroups and interactions, handle missing data, and account for the sampling strategy.

Response: We have added more detail to the areas that have been requested.

RESULTS: The manuscript might flow better if it were structured as follows:

Participants:

(a) There is a need to report the numbers of individuals at each stage of the study, including eligibility, inclusion, follow-up, and analysis stages. This is a major need for improvement as it provides transparency and allows readers to understand the sample size and potential biases.

(b) The reasons for non-participation at each stage should be provided. This is a minor need for improvement as it helps identify potential sources of bias or limitations in the study.

Descriptive data:

(a) The characteristics of study participants, such as demographic, clinical, and social information, should be reported.

(b) The number of participants with missing data for each variable of interest should be indicated.

Response: We have provided all the information that is available in relation to the participant information you have requested. This whole section has been reworded in response to similar comments from other reviewers.

Main results:

(a) Unadjusted estimates are provided, but the report should also include confounder-adjusted estimates and their precision. This is a major need for improvement as it allows for a more robust analysis and interpretation of the results.

Response: We have reported the adjusted and unadjusted results in Table 2.

(b) Category boundaries should be reported when continuous variables are categorized. However, since no continuous variables are mentioned in the given information, this is not applicable.

(c) The translation of estimates into absolute risk for a meaningful time period should be considered if relevant. It is not mentioned in the given information.

Other analyses:

The sensitivity analysis is reported, which is a positive aspect of the study.

Reviewer: 3

Dr. Tori L Cowger , Harvard T.H. Chan School of Public Health

Comments to the Author:

Thank you for the opportunity to review your manuscript. I have outlined my comments and recommendations below, including major comments that I believe would need to be addressed before publication and minor comments that might improve clarity and strengthen this work.

MAJOR COMMENTS:

1. Sampling frame, selection, and target population

The authors report that their sample was selected via an internet survey among a non-probability sample of 5,000 participants recruited from a pool of participants interested in completing market research surveys and through recruitment via advertising on various online platforms. Authors also state that this survey was conducted as a “wider piece of work” (page 5 of 23, line 52), however, they do not state the objective or aims of primary study for which this sample was recruited. Authors describe quotas based on strata of demographic factors including age, working status, and social grade, however, they do not state whether the population sampled met these recruitment targets nor do they present any descriptive statistics of the full study population compared to their target population. This information is essential in interpreting the results of this survey, especially since it seems likely that the population of individuals who are interested and available to complete market research surveys is likely to differ in key ways from the general population (e.g., access to resources, health information, education, etc.) that are likely to impact whether or not a parent sends a child to school while symptomatic.

Response: We have provided more information about the wider study and the aims, sampling, and participant recruitment.

In addition, authors state that of their 4,962 participants, 941 had at least one school-aged child, with 251 of these participants reporting that their child had stay-at-home symptoms and were therefore included in the primary analysis. This is a relatively small subset of the survey population, and it would be helpful to know how the population included in the analysis compares to both the target population and other subsets of the study population (e.g., parents who had children that did not report symptoms).

Response: We have reported that parents (n=941) reported on symptom(s) for 1,533 children aged four to 17 years. 48% of parents (n=454, 95% CI 45% to 51%) reported that at least one of their children had experienced at least one symptom and 27% (n=251, 95% CI 24% to 30%) reported that at least one child had experienced at least one stay-at-home symptom. The number of participants used in this sensitivity analysis differ because not all the children in the sample had attended school while symptomatic. This clarification has been added to the manuscript.

On page 12 of 23, authors acknowledge non-probability sampling as a limitation, however they go on to say that they “have no reason to suspect that the associations within the data cannot be generalized.” I have two key concerns related to this statement, as a highly selected sample presents threats to both internal and external validity.

First, the authors acknowledge some differences between their study population and their target population and it seems like the final sample may not be representative of the general population – For example, e.g., nearly half n=123/151 of parents of symptomatic children have a degree or higher education levels, which is substantially higher than reported in UK census data (~33%). In addition, authors report that they could not report results by race/ethnicity as more than 90% of the study population were of white ethnicity. The over-representation of certain groups in the study sample has the potential to bias the estimate of prevalence of school attendance while symptomatic if these characteristics associated with selection into the sample are associated with symptomatic school attendance. In addition, differences between sample and target population can represent threats to internal validity and bias estimates of association if sample selection is associated with both the exposures of interest in the study (i.e., characteristics presented in table 1), and their outcome, symptomatic school attendance. It seems likely that characteristics associated with selection into the sample (e.g., parental education) would be associated with both perceptions of illness and also associated to sending children to school when sick. It would be helpful to understand the magnitude of this potential bias by comparing the characteristics of the study population those available in the target population. I would suggest that where possible, authors provide a table comparing the sociodemographics of their study population with that of their target population where publicly available data are available. In addition, it would be helpful to know how the final sample in this analysis compares to the full population sample and relevant subsets. To address this, authors could present a supplementary table comparing the demographic characteristics they currently present in table 1 for the full study population (~N=5000), the population of all parents in the full sample (n=941), and parents of children with symptoms to get a better sense of how similar or different these populations are. This may help give a better sense of the magnitude of potential bias that might be introduced during study recruitment and through exclusions.

Response: We have provided more details about the participants and response rates in the manuscript and in response to previous comments. However, in response to your comment and for ease we have repeated: Recruitment followed a complex pattern, in which responders from earlier waves were invited to participate and additional respondents from the panel were then invited to take part to make up for non-respondents. Importantly, respondents were drawn from a panel that was originally self-selected. Providing a response rate therefore gives no information as to the level of non-response bias present in our sample as compared to the general population. In addition, quota samples aim to minimise response bias by filling predetermined targets so that the social and personal characteristics of the participants match those of the national population. As such, participants who belong to a quota that has already been met are prevented from completing the survey. Therefore, response rates are not useful indicators of response bias in quota samples and are not usually reported. We have updated the manuscript to include this rationale.

Second, the authors mention the “wider population of parents in the UK” which is presumably their target population, and seem to be discussing the internal validity of their results, but use the term “generalize” which is more appropriately applied when discussing external validity (e.g., populations of parents outside of the UK). Authors should consider being more precise in their language.

Response: We have made it clear that our study is reporting about parents and therefore aims to be generalisable to UK parents with school-aged children and not the wider sample.

2. Framing and language of “Engaging in risky behavior”

Throughout their paper, the authors refer to children who attend school or activities outside of the household as “engaging in risky behavior.” This framing and language seems vague and also has the potential to stigmatize children and their parents. I would suggest authors consider using different language that is more specific to the outcome the authors measured, for example, “attending school while symptomatic,” “socializing while symptomatic” or “school-based presenteeism,” which authors refer to in their introduction.

Response: Presenteeism doesn't usually include behaviours that relate to socialising rather focuses on where the person is when symptomatic (e.g., work or school). Socialising while symptomatic is also inaccurate as children may have attended school or a club outside of school, although not socialised with anyone. However, we agree that this may be stigmatising and not reflective of the outcome that we measured. Therefore, we have reframed our wording to non-adherent behaviour.

3. Missing results of sensitivity analyses.

Authors mention a sensitivity analysis wherein they considered the outcome of school attendance only (vs. their primary analysis which considered school attendance together with attending other activities), however, the results of these analysis are not presented in detail anywhere. On page 8 of 23, the authors state that n=227 parents were included in this analysis, but do not explain why their sample size is different for the sensitivity analysis vs. their main analysis. The authors also note several key differences between primary and sensitivity analyses, but without seeing the full results of the sensitivity analyses presented, these results are hard to interpret. Authors should consider presenting the results of the sensitivity analysis in full in their supplementary material and should explain why the sample sizes are different between primary and sensitivity analyses.

Response: We have described that the sensitivity analysis were conducted to identify any differences between our findings and if we only measured attending school while symptomatic rather than also including the variables that related to other activities and socialising. This was to limit the bias that may have occurred with the attitudes that we measured in connection to school attendance. However, we note that “school attendance was not associated with parent age, but was associated with parents agreeing that their child did not want to take time off school. No other differences in associations were found between school attendance and non-adherent behaviour and our predictor variables.”

Minor Comments:

1. Descriptive statistics of stay-at-home symptoms: Authors list out 14 symptoms included in the survey on page 19 of 23 and list out several as “stay at home” symptoms per UK guidelines on page 7 of 23. On page 8 of their results section, authors report that 48% of parents (n=454/941) reported their child reported at least one symptom, but only 27% (n=251) reported a stay-at-home symptom. This seems like a big difference to me, and it may be helpful to include a supplemental table that delineates which of the symptoms surveyed were considered ‘stay-at-home’ symptoms per study criteria and which symptoms were most common among study participants reporting symptoms. This

would help readers better understand the study population included in the sample and symptoms experienced by those included and excluded from the study.

Response: We agree that this may have been useful. However, we asked parents to report the symptoms for each of their children (up to four) and then to report about one child. For households that had children who experienced multiple experiences of symptoms we are unable to determine which child the parent reported about in later questions. Therefore, a breakdown of the symptoms in this way would not report the data that you have requested.

2. Description of symptom/symptoms: it would be helpful to know which and how many symptoms children included in the study were experiencing and whether that was related to school attendance while symptomatic. For example, are children with certain sets of symptoms (e.g., cough only) more likely to be sent to school than children with different symptoms? I suggest that the authors add both stay-at-home symptom type and number of stay-at-home symptoms to Table 1.

Response: Please see the response to the above comment as the same reasoning applies.

3. Power and statistical significance: the final sample size included in the study is relatively small, which likely impacts the statistical power to detect differences between groups in the study. However, authors do not mention this as a limitation, and there are several instances where authors interpret lack of statistical significance as an absence of a true association rather than lack of statistical power. For example, on page 12 of 23, authors describe a null association between perceptions of being behind in school and attending school while symptomatic and interpret this as contrasting with prior literature. However, this relationship was statistically significant in univariate analyses, with similar point estimates in both unadjusted and adjusted analyses. Therefore, this seems likely an issue of statistical power rather than truly conflicting with prior research, and authors should consider modifying language to reflect this.

Response: We have added to the discussion that the study had a relatively small sample size and therefore our findings may have been due to this rather than no effects.

4. Multiple testing: The authors report results of many statistical tests in their primary analyses and presumably conducted additional statistical tests in their supplementary analyses. However, I didn't see if the authors mentioned their approach or rationale for multiple testing?

Response: We report the rationale for multiple testing, primarily to control for confounders and second for the sensitivity analysis, which we have responded to above.

VERSION 2 – REVIEW

REVIEWER	Most, Zachary M Children's Health System of Texas
REVIEW RETURNED	10-Aug-2023

GENERAL COMMENTS	Thank you to the authors for your thoughtful revisions to this manuscript. It is much improved and could be ready for publication in its current state. The title could be slightly re-worked to make it more clear. Currently, the three items listed: "attend school" "other activities", or "socialise" is written in a way that implies that the word "attend" should be applied before "other activities" and "socialise", but that would not make sense before "socialise". Potential alternates could be:
--

	"Why do children attend school, engage in other activities, or socialise when they have symptoms of an infectious illness? A cross-sectional survey" "Why do children attend activities outside the home or socialise inside the home when they have symptoms of an infectious illness? A cross-sectional survey" The abstract is excellent. The introduction is thorough and much improved. The methods is section is much improved. Thank you for providing more detail on how the sample was recruited. The results section is very good. The comment in the footnote to Table 1 "*Column totals do not add to 100% as participants could tick multiple options" could be edited or removed. The percentages shown in the table appear to be for row totals and columns will not sum to 100% when row totals are presented. I think what the statement was trying to say was something along the lines of "The column totals do not add to 84 as participants could tick multiple options" (since the total number of non-adherent presented in the text and Table 1 is 84). The discussion section is also very strong now. For the discussion on the fact that there was no association between child non-adherence and parents who felt their child had missed too much school or were behind in school, you could make note that the aOR for "My child is behind at school is 0.84 (95% CI 0.69 to 1.02), which is a trend towards the association you expected and perhaps the reason you did not find an association in your study was because the estimates were too imprecise/sample size was not large enough. Thank you for taking the time to do these revisions.
--	---

REVIEWER	Cowger , Tori L Harvard T.H. Chan School of Public Health
REVIEW RETURNED	24-Aug-2023

GENERAL COMMENTS	Thank you for the opportunity to review your manuscript. I believe the revisions the authors made have substantially improved the manuscript. I only have a few outstanding minor questions/clarifications that would further strengthen this work, outlined below: 1. Additional details of participant characteristics selected into the sample and discussion of any implications On page 5 of the revised manuscript, the authors have added details about how the population of parents included in the study compares to the target population (parents of dependent children in the UK) with respect to employment and income. These additions are very helpful in understanding the degree to which selection into the study population may have influenced both prevalence estimates and associations reported in the study. In addition to income and employment, the authors also measured and report parent education level in Table 1, however, they do not provide education statistics for their target population, parents in the UK, as they do for income and employment. In the study
--

	sample, nearly half n=123/151 of parents of symptomatic children have a degree or higher education levels, which is substantially higher than reported in UK census data (~33%). It would be helpful for authors to add a similar comparison for parental education since it seems likely that parental education would be associated with both perceptions of illness and also associated to sending children to school when sick. If the levels of education in the study sample differ substantially from the target population, then the authors might also consider adding a sentence to the discussion section about the anticipated direction of bias, and implications for interpreting their study results. 2. Clarification and addition of results for sensitivity analyses Authors describe a sensitivity analysis wherein they considered the outcome of school attendance only (vs. their primary analysis which considered school attendance together with attending other activities). In response to my comments on the initial draft, authors also provide a clarification as to why the sample size was different for the sensitivity analysis considering school attendance only (n=227) as compared to their primary analysis (n=251), which considered school attendance and activities outside of school. On page 8 of the revised manuscript, authors added the clarification, “Fewer participants were included in the sensitivity analysis because not all 251 children attended school while symptomatic with stay-at-home symptoms.” I may have misunderstood, but even after this clarification, I was still unsure of why the sample size of the main and sensitivity analyses were different. My understanding from the authors description is that the difference between the main and sensitivity analyses was the classification of the outcome of “non-adherent” (i.e., in the main analysis non-adherence was defined as attending school and/or other activities (shown in Table 2) while symptomatic whereas in the sensitivity analysis “non-adherence” was based on school attendance only. Could the authors clarify that this understanding is correct? If so, it remains a bit unclear to me why the sample size for the sensitivity analysis should be different given that in Table 2, authors show the distribution of school attendance for the full sample of n=251 participants. Additional clarity on the sensitivity analyses would be very helpful, and I strongly recommend that authors add the results of these analysis to the supplemental appendix, in addition to the brief mention of the results in the text. 3. Date in the abstract The abstract states “since September of 2019,” but I believe this should be “September 2021”? It might also be helpful to add when the study was conducted/approximately how many months of the school year to interpret the prevalence estimate authors present in the abstract (e.g., “Over the first XX months of the school year, one-third of children...”
--	---

VERSION 2 – AUTHOR RESPONSE

Reviewer: 1
Dr. Zachary M Most, Children’s Health System of Texas
Comments to the Author:

Thank you to the authors for your thoughtful revisions to this manuscript. It is much improved and could be ready for publication in its current state.

The title could be slightly re-worked to make it more clear. Currently, the three items listed: "attend school" "other activities", or "socialise" is written in a way that implies that the word "attend" should be applied before "other activities" and "socialise", but that would not make sense before "socialise". Potential alternates could be:

"Why do children attend school, engage in other activities, or socialise when they have symptoms of an infectious illness? A cross-sectional survey" [NOTE FROM THE EDITORS: This is our preferred suggestion]

"Why do children attend activities outside the home or socialise inside the home when they have symptoms of an infectious illness? A cross-sectional survey"

Response: Thank you for the title suggestions we have changed our manuscript title to “Why do children attend school, engage in other activities, or socialise when they have symptoms of an infectious illness? A cross-sectional survey.”

The abstract is excellent.

The introduction is thorough and much improved.

The methods section is much improved. Thank you for providing more detail on how the sample was recruited.

The results section is very good. The comment in the footnote to Table 1 "Column totals do not add to 100% as participants could tick multiple options" could be edited or removed. The percentages shown in the table appear to be for row totals and columns will not sum to 100% when row totals are presented. I think what the statement was trying to say was something along the lines of "The column totals do not add to 84 as participants could tick multiple options" (since the total number of non-adherent presented in the text and Table 1 is 84).

Response: Thank you for the suggestion, we have removed the footnote.

The discussion section is also very strong now. For the discussion on the fact that there was no association between child non-adherence and parents who felt their child had missed too much school or were behind in school, you could make note that the aOR for "My child is behind at school is 0.84 (95% CI 0.69 to 1.02), which is a trend towards the association you expected and perhaps the reason you did not find an association in your study was because the estimates were too imprecise/sample size was not large enough.

Response: Thank you for the comment. We have added this rationale for why we did not find a significant association for this result to our discussion.

Thank you for taking the time to do these revisions.

Reviewer: 3

Dr. Tori L Cowger , Harvard T.H. Chan School of Public Health

Comments to the Author:

Thank you for the opportunity to review your manuscript. I believe the revisions the authors made have substantially improved the manuscript. I only have a few outstanding minor questions/clarifications that would further strengthen this work, outlined below:

1. Additional details of participant characteristics selected into the sample and discussion of any implications

On page 5 of the revised manuscript, the authors have added details about how the population of parents included in the study compares to the target population (parents of dependent children in the UK) with respect to employment and income. These additions are very helpful in understanding the degree to which selection into the study population may have influenced both prevalence estimates and associations reported in the study. In addition to income and employment, the authors also measured and report parent education level in Table 1, however, they do not provide education statistics for their target population, parents in the UK, as they do for income and employment. In the study sample, nearly half $n=123/151$ of parents of symptomatic children have a degree or higher education levels, which is substantially higher than reported in UK census data (~33%). It would be helpful for authors to add a similar comparison for parental education since it seems likely that parental education would be associated with both perceptions of illness and also associated to sending children to school when sick. If the levels of education in the study sample differ substantially from the target population, then the authors might also consider adding a sentence to the discussion section about the anticipated direction of bias, and implications for interpreting their study results.

Response: The current census data do not report the education levels for people with and without school-aged children and we were unable to find these data elsewhere. However, we have reported these data in the manuscript and discussed why this may have had an impact on our findings.

2. Clarification and addition of results for sensitivity analyses

Authors describe a sensitivity analysis wherein they considered the outcome of school attendance only (vs. their primary analysis which considered school attendance together with attending other activities). In response to my comments on the initial draft, authors also provide a clarification as to why the sample size was different for the sensitivity analysis considering school attendance only ($n=227$) as compared to their primary analysis ($n=251$), which considered school attendance and activities outside of school. On page 8 of the revised manuscript, authors added the clarification, "Fewer participants were included in the sensitivity analysis because not all 251 children attended school while symptomatic with stay-at-home symptoms." I may have misunderstood, but even after this clarification, I was still unsure of why the sample size of the main and sensitivity analyses were different. My understanding from the authors description is that the difference between the main and sensitivity analyses was the classification of the outcome of "non-adherent" (i.e., in the main analysis non-adherence was defined as attending school and/or other activities (shown in Table 2) while symptomatic whereas in the sensitivity analysis "non-adherence" was based on school attendance only. Could the authors clarify that this understanding is correct? If so, it remains a bit unclear to me why the sample size for the sensitivity analysis should be different given that in Table 2, authors show the distribution of school attendance for the full sample of $n=251$ participants. Additional clarity on the sensitivity analyses would be very helpful, and I strongly recommend that authors add the results of these analysis to the supplemental appendix, in addition to the brief mention of the results in the text.

Response: Thank you for your comment and you have interpreted our manuscript correctly. The numbers are different as only 227 children attended school while symptomatic while 251 children attended school or other activity outside of school. Thus, 24 children only attended other activities while they were symptomatic and were adherent to not attending school, which is why we ran this sensitivity analysis. We wanted to identify whether removing these 24 children would impact our results. However, there were minimal differences. Of course, we are happy to take the editor's view as to whether the sensitivity analysis should be included as supplementary material, although we have discussed the relevant findings in our manuscript and suggest that this is sufficient.

3. Date in the abstract

The abstract states "since September of 2019," but I believe this should be "September 2021"? It might also be helpful to add when the study was conducted/approximately how many months of the school year to interpret the prevalence estimate authors present in the abstract (e.g., "Over the first XX months of the school year, one-third of children...")

Response: Thank you for highlighting this error. We have changed this to 2021. The design section reports that data were collected between 19 November and 18 December 2021. Therefore, parents were reporting on up to four months of the school year.

*** **

COI statements:

Reviewer: 1

Competing interests of Reviewer: n/a.

Reviewer: 3

Competing interests of Reviewer: None.